# Go big or go home: A model-based assessment of general strategies to slow the spread of forest pests via infested firewood

**Peter C. Jentsch**[1,2,3]*, **Chris T. Bauch**[1], **Denys Yemshanov**[2], **Madhur Anand**[3]

**1** Department of Applied Mathematics, University of Waterloo, Waterloo, Ontario, Canada, **2** Natural Resources Canada, Canadian Forest Service, Great Lakes Forestry Centre, Sault Ste. Marie, Ontario, Canada, **3** School of Environmental Sciences, University of Guelph, Guelph, Ontario, Canada

* pjentsch@uwaterloo.ca

**Data Availability Statement:** All data files and code are available at the following URL: https://git.uwaterloo.ca/pjentsch/plos_firewood_model.

**Funding:** This work has been funded by Canadian Forest Service - Canadian Food Inspection Agency

## Abstract

Invasive pests, such as emerald ash borer or Asian longhorn beetle, have been responsible for unprecedented ecological and economic damage in eastern North America. These and other wood-boring invasive insects can spread to new areas through human transport of untreated firewood. Behaviour, such as transport of firewood, is affected not only by immediate material benefits and costs, but also by social forces. Potential approaches to reduce the spread of wood-boring pests through firewood include raising awareness of the problem and increasing the social costs of the damages incurred by transporting firewood. In order to evaluate the efficacy of these measures, we create a coupled social-ecological model of firewood transport, pest spread, and social dynamics, on a geographical network of camper travel between recreational destinations. We also evaluate interventions aimed to slow the spread of invasive pests with untreated firewood, such as inspections at checkpoints to stop the movement of transported firewood and quarantine of high-risk locations. We find that public information and awareness programs can be effective only if the rate of spread of the pest between and within forested areas is slow. Direct intervention via inspections at checkpoints can only be successful if a high proportion of the infested firewood is intercepted. Patch quarantine is only effective if sufficiently many locations can be included in the quarantine and if the quarantine begins early. Our results indicate that the current, relatively low levels of public outreach activities and lack of adequate funding are likely to render inspections, quarantine and public outreach efforts ineffective.

## Introduction

Invasive species pose a significant economic and ecological threat to Canada's forest ecosystems [1, 2]. In North America, significant funding has been allocated by federal, state and provincial agencies for large-scale control programs to prevent or mitigate these damages with mixed success [3, 4]. Controlling the spread of invasive pests can be difficult because the long-distance spread of invasive organisms is often assisted by human activities [1, 5]. For example,

Forest Invasive Alien Species Initiative, https://www.nrcan.gc.ca/our-natural-resources/forests-forestry/13497, the Ministry of Natural Resources, and NSERC Discovery Grants on behalf of Madhur Anand and Chris T. Bauch, respectively. The funders had no role in study design, data collection and analysis, decision to publish, or preparation of the manuscript. There was no additional external funding received for this study.

**Competing interests:** The authors have declared that no competing interests exist.

introduction and spread of Emerald ash borer, a harmful forest pest in the North America [6–8] has been attributed to human factors, such as vehicle transport [9] and recreational travel [10].

The growing problem of invasive species is broadly associated with human mobility, including recreational travel [1, 5, 11, 12]. Outdoor recreation is widespread in North America, and the extent of recreational activities is expected to increase [13–15]. In North America, national, provincial and state parks, national forests, and state and Crown lands are common destinations for recreational activities [16, 17]. In Canada, recreational activities, especially camper travel, often take place in forested areas and may contribute to spread of harmful invasive pests. In particular, the movement of untreated firewood by campers has been widely acknowledged as a potential introduction pathway for invasive forest pests [2, 10, 18–20]. Movement of untreated firewood has been linked to the spread of two harmful wood-boring pests, the Asian longhorned beetle (*Anoplophora glabripennis* Motschulsky) and the emerald ash borer (*Agrilus plannipennis* Fairmaire), in the United States and Canada (USaC) [21, 22].

Firewood is often moved to distant locations by campers for recreational purposes [2, 23]. For example, Haack et al. (2010) has found live bark- and wood-boring insects in 23% of the firewood pieces, surrendered at the checkpoint station at Mackinac bridge connecting Michigan's Lower and Upper Peninsulas and an additional 41% had signs of prior borer infestation. Jacobi et al. (2011) reported the emergence of live insects from 47% of the firewood bundles purchased from various US retailers. To reduce the risk of future pest infestations, USaC have implemented various regulations on movement of untreated firewood, including bans for out-of-province movement of untreated firewood and restrictions for its transport by short distances [19, 23–25]. Also a number of public outreach campaigns have been undertaken to educate the general public about the threats associated with the movement of untreated firewood and its potential to spread harmful invasive pests. Several strategies have been developed to prevent (or minimize) the movement of firewood with recreational travel, including outreach campaigns in public media, enforcements with the inspections at check points for transported firewood, and area quarantine with the restrictions on firewood movement from/to the area of concern. In particular, public outreach campaigns have become widespread with significant funding by local, municipal, and provincial governments on measures such as advertisements along major highways and in public media and educational information in websites and printed media. The use of enforcement and quarantine options is less common but is gaining acceptance as a last resort measure and was implemented at least a few times over the past decade, to varying degrees of success [10, 18, 25].

Assessing the efficacy of the measures aimed to prevent the movement of firewood with recreational travel is a daunting task. Outreach campaigns may spread information widely but there is no guarantee that campers will pay attention and comply with the firewood restriction warnings. Many outreach activities (such as posting ads in public media or distributing flyers) are often implemented sporadically at local scales using local municipal and provincial budgets [19], which makes the assessment of their efficiency difficult. These activities may simultaneously occur in different places and times with little or no coordination, and are difficult to track in time and space.

Alternatively, the enforcement options (such as quarantine or checkpoint inspections for illegal movement of firewood) are gaining acceptance and may be perceived as more effective localized means to stop the movement of untreated firewood by campers. Nevertheless, assessing the effectiveness of enforcement actions is challenging due to a very small scale of enforcement actions (often implemented by individual states or provinces at selected locations) and lack of compliance data.

Mechanistic models of forest invasions have been studied for decades [26, 27], but explicit modelling and consideration of human factors, and the feedback between humans and the environment is relatively new. Ali et al. and Barlow et al. [18, 28] proposed two models of forest pest spread through firewood transport. The first study presented a differential equation model, and the second an agent-based model, both assuming that humans are the primary long-distance movers of forest pests. The models proposed in [18, 28] coupled infestation dynamics with the social dynamics. However both studies considered a small and idealized spatial structure: two patches in Barlow's et al. [18] study and ten patches in Ali's et al. model [28]. Often, illegal movement of firewood occurs over large distances and may involve visits to multiple recreational destinations that are connected differently to one another.

In this study we consider movement of infested firewood to multiple recreational destinations over a complex recreational travel network. We explore the efficacy of common measures aimed to stop the movement of untreated firewood by recreational travelers. To accomplish this, we propose a mechanistic differential equation model that combines human-mediated movement of forest pests through a camper travel network that includes nonlinear feedbacks from social factors, such as population response to strategies preventing the movement of untreated firewood. We identify three basic methods to stop or slow the spread of invasive pests by transport of infested firewood: public awareness campaigns, direct interception of transported firewood at checkpoints near recreational destinations, and quarantining recreational destination sites for movement of firewood. While the first option is more common, the latter has been implemented seldom over the past decade due to legal and liability constraints [24, 29–31]. We implement the options for intercepting the movement of firewood to slow the spread of invasive pests in a mechanistic metapopulation model, and use the replicator equation to represent social learning dynamics (see [18, 32–34]). We also evaluate local quarantine at recreational destinations as an alternative control method. Quarantine means closing the site to visitors for a length of time, in order to reduce the amount of transported firewood and slow spread of invasive organisms from other infested locations. Our implementation of quarantine measures follow common practices aimed to slow the spread of invasive species (such as the spread of emerald ash borer in USaC [35, 36]). We apply our mechanistic model to explore the effectiveness of these control measures to slow the spread of an idealized wood-boring invasive pest moved to a set of recreational destinations by recreational travelers transporting untreated firewood. We apply the model to a network of provincial parks and campgrounds in three provinces of central Canada—Manitoba, Ontario, and Quebec.

## Materials and methods

We consider a landscape of $N$ patches, where a patch is represented as $i \in [1, N]$. Each patch represents a recreational destination (eg. provincial parks and campgrounds) with associated neighbouring human population centres. Each patch undergoes its own internal pest and social dynamics. We describe the spread of an invasive pest with the movement of firewood through the network of N patches with a mechanistic metapopulation model based on [18] that captures the spread of an infestation between the patches. The advantage to metapopulation models in this context is suitability for capturing dynamics of a highly fragmented population spread over a broad geographic region. Using the data documenting reservations of provincial campgrounds in Ontario, Manitoba and Quebec ([37], we created a graph of camper travels which depicts a spatial travel network between origin locations (which correspond to residential addresses of camper travelers) and recreational destinations (campgrounds in provincial parks and historic sites). The camper travel network is described by a graph with coefficients $P_{i,j}$ denoting the relative frequency of camper movements between

origin locations $j$ and recreational destination locations $i$ (see more details on spatial data in section). Specifically, for a given location $j$, $P_{i,j}$ is the fraction of trips that go from $j$ to $i$ each year, so we have $\sum_{i=1}^{N} P_{i,j} = 1$. Consider a patch $i$ with an enforcement intervention, such as firewood movement quarantine, or a voluntary firewood surrender checkpoint aimed to stop the flow of untreated firewood from that location. We denote $C_e$ as the percentage of infested firewood that can be intercepted on a route between two locations $i$ and $j$, $0 \leq C_e \leq 1$. Interception at $i$ may reduce the movement of infested firewood from a patch $i$ to other patches $j$, so $C_e$ indicates, in relative terms, the magnitude of interception efforts.

We also consider a public outreach campaign that can take place at a patch $i$. It is common that only a portion of campers visiting a patch $i$ may be aware of and decide to comply with the public outreach message. We model the social awareness campaign as an increase of the net social cost of transporting firewood. We further conduct sensitivity analyses to compare the efficacy of enforcement vs. outreach measures aimed to stop the movement of firewood and reduce the rates of infestation.

## Pest spread model

We begin with defining the equation for a population of susceptible host trees that may be attacked by an invasive pest. The pest can be introduced though untreated infested firewood. Variables, their interpretations, and corresponding baseline ranges are shown in Table 1. We assume that a tree population that is susceptible to pest attack undergoes logistic growth in the absence of infestation to a carrying capacity $K$. The population of susceptible trees, $S_i(t)$, at a

**Table 1. Parameters and default values.**

| Name | Default Value, (Range explored) | Units | Interpretation |
|------|--------------------------------|-------|----------------|
| $N$ | 2250 | Patches | Number of patches in the network |
| $S_i$ | Site specific | Trees | Number of susceptible trees in patch $i$ |
| $I_i$ | Site specific | Trees | Number of infested trees in patch $i$ |
| $B_i$ | Site specific | Trees | Infested firewood in patch $i$ |
| $L_i$ | Site specific | Unitless | Fraction of local strategists in patch $i$ |
| $r$ | 0.02, [0.01, 0.06] | New trees per tree per year | Tree growth rate |
| $A$ | 0.001, [0.00065, 0.0014] | Number of infested trees per susceptible-infested contact per year | Transmission rate of pest |
| $\gamma$ | 1.4, [0.8, 1.8] | Trees per year | Decay rate for infested trees |
| $K$ | 5000 | Trees | Carrying capacity of each patch |
| $U$ | 0, [-5, 5] | Utility | Social cost to transport firewood, or incentive to buy locally |
| $C_e$ | 0, [0.0, 1.0] | Unitless | Interception fraction |
| $f$ | 0.1, [0.01, 0.13] | Utility per capita | Impact of local infection on strategy |
| $s$ | 0.1 | Utility per capita | Strength of social norms |
| $\sigma$ | 0.1 | Strategy changes per capita per year | Rate of social learning |
| $P_{i,j}$ | See below | Unitless | Fraction of trips that go from $j$ to $i$ each year. |
| $d$ | 0.1 [0.05, 0.3] | Logs per year | Rate of transmission of infested firewood between patches |
| $I_a$ | 1 [0.5, 5] | Trees | Value at which transmission rate of pest is halved due to density dependence |
| $k$ | 1 | Unitless | Steepness of sigmoid function |
| $V$ | Empty, [0 patches, 500 patches] | Patches | Set of patches to be quarantined |
| $\Delta t$ | 0, [0, 5] | Years | Length of quarantine |
| $t_0$ | 0, [0, 5] | Years | Time between initial infestation and patch quarantine |

patch $i$ is being infested from firewood arriving with campers at $i$ at a rate $A$:

$$\frac{dS_i}{dt} = \underbrace{rS_i\left(1 - \frac{(S_i + I_i)}{K}\right)}_{\text{Logistic Growth Of Forest}} - \underbrace{AS_i(I_i + B_i)\theta_k(I_i - I_a)}_{\text{Infestation term}} \tag{1}$$

where $\theta_k(I_i)$ is a sigmoid function such as

$$\theta_k(x) = \frac{1}{1 + e^{-kx}} \tag{2}$$

Terms $S_i$ and $I_i$ are the number of susceptible and infected trees, respectively, at patch $i$. $B_i$ is the quantity of infested firewood in patch $i$, which we assume has the same probability of pest transmission within patch as infested trees. We choose the carrying capacity $K$ to be the same in each patch for simplicity. The term $AS_iI_i\theta_k(I_i - I_a)$ represents intra-patch infestation with a density dependent population, parameterized by $k$ and $I_a$, where $I_a$ determines population of infested trees at which transmission is halved, and $k$ is is a constant which affects the sharpness of the transition of $\theta_k(x)$ at $I_a$. We assume that there is an influx of pest organisms entering a patch $i$ with firewood which defines the propagule pressure at $i$. Infested trees at $i$ are assumed to die at a constant rate $\gamma$, giving the following equation for the infested tree population of a patch.

$$\frac{dI_i}{dt} = \underbrace{-\gamma I_i}_{\text{Death of infested trees}} + \underbrace{AS_i(I_i + B_i)\theta_k(I_i - I_a)}_{\text{Susceptible become infested}} \tag{3}$$

The patches are spatially coupled through the transport of firewood by recreational travelers. The infestation rate at $i$ depends on the number of visitors transporting infested firewood to $i$, which is also a function of the social dynamics at $i$, such as the enforcement, or public outreach measures described by a utility function, presented in [33], and applied to forest modelling in [18, 38]. Let $L_i$ be the proportion of visitors to patch $i$ who do not transport firewood and buy it locally, and $d$ rate of exportation of infested logs. The rate of infested wood coming into patch $i$ can be estimated as:

$$d \sum_{j=1, j \neq i}^{N} P_{i,j}(1 - C_e)(1 - L_j)I_j$$

The dynamics of $L_i$ (the number of local transporters in patch $i$), is modelled by a replicator dynamics model that is suitable for describing systems where social learning occurs [33, 34], and is described in the section below.

## Social dynamics model

We model the proportion of visitors who choose to use local firewood, $L_i$ as a function of both the perceived threat of introduced pests, and the social cost of illegally transporting infested firewood. We refer to visitors who choose to use local firewood as local strategists, and visitors who do not use local firewood as transport strategists hereafter. Let $C_t$ be the cost of transporting firewood and $C_l$ the cost to obtain it locally (and therefore avoid moving invasive pests to a patch $i$). We adopt the social influence model from [18], which is based on models of [33] and [34], which we will summarize below. We define the social utilities corresponding to the

strategies of transporting firewood ($P_t$) and buying it locally ($P_l$) as

$$P_t = -C_t + s(0.5 - L_i) - fI_i$$

$$P_l = -C_l + s(L_i - 0.5)$$

Transportation becomes a less attractive strategy if infestation is more prevalent, depending on the size of $f$. The parameter $f$ controls the extent to which a local infestation causes behaviour change in that population. The parameter $s$ controls the degree to which individuals are influenced by the the majority opinion in their patch (i.e. peer pressure). We assume that both local strategists and transport strategists in a patch $i$, given by $L_i$ and $1 - L_i$ respectively, decide whether to change their strategy at the same rate, $\sigma$. Their decision is made by considering which strategy will maximize their utility $P_l - P_t$ at that point, leading to the following expression for the rate of change of the local strategist population:

$$\frac{dL_i}{dt} = \sigma L_i (1 - L_i)(P_l - P_t)$$

We replace the individual costs of $C_t$, $C_l$ with the net utility value $U = C_t - C_l$. The cost difference $U$ abstracts from the explicit definition of costs of using firewood [18] and allows including exogenous social incentives and motivation, such as awareness about the problem or any other form of social influence from outside each location $i$. A term $B_i$ is introduced to represent the amount of local firewood available in patch $i$. For simplicity, we assume that the tree mortality rate at a patch $i$ is only caused by infestation, so the mortality rate is the same as the death rate of the infested trees

$$\frac{dB_i}{dt} = \underbrace{-\gamma B_i}_{\text{Decay of fallen wood}} + \underbrace{d \sum_{j=1, j \neq i}^{N} P_{i,j}(1 - C_e)(1 - L_j)I_j}_{\text{Import of fallen wood}} \tag{4}$$

Because the infested wood imported into patch $i$ in Eq 4 must come from another patch in the system, we subtract the corresponding term for leaving wood, $d \sum_{j=1, j \neq i}^{N} P_{j,i}(1 - C_e)(1 - L_i)I_i$ from Eq 6 which describes the rate of change of infested population in a patch $i$. Using the notation in Eqs (5), (6), (7) and (8), we formulate the problem of buying firewood locally vs. transporting it from other potentially infested locations as follows:

$$\frac{dS_i}{dt} = \underbrace{rS_i \left(1 - \frac{(S_i + I_i)}{K}\right)}_{\text{Logistic Growth Of Forest}} - \underbrace{AS_i(I_i + B_i)\theta_k(I_i - I_a)}_{\text{Infestation term}} \tag{5}$$

$$\frac{dI_i}{dt} = \underbrace{-\gamma I_i}_{\text{Death of infested trees}} + \underbrace{AS_i(I_i + B_i)\theta_k(I_i - I_a)}_{\text{Susceptibles become infested}} - \underbrace{d \sum_{j=1, j \neq i}^{N} P_{j,i}(1 - C_e)(1 - L_i)I_i}_{\text{Total infested wood leaving due to transport}} \tag{6}$$

$$\frac{dB_i}{dt} = \underbrace{-\gamma B_i}_{\text{Decay of firewood}} + \underbrace{d \sum_{j=1, j \neq i}^{N} P_{i,j}(1 - C_e)(1 - L_j)I_j}_{\text{Import of fallen wood}} \tag{7}$$

$$\frac{dL_i}{dt} = \sigma L_i(1 - L_i)(\underbrace{U}_{\text{Net cost to transport firewood}} + \underbrace{s(2L_i - 1)}_{\text{Social influence term}} + \underbrace{fI_i}_{\text{Impact of infestation}}) \qquad (8)$$

Table 1 lists the model notation.

## Patch-quarantine strategies

Let $V \subset [1, N]$ be a set of patches under a quarantine. We use the patches (nodes of the camper travel network) with the largest (shortest-path) betweenness centrality [39, 40], which is a common approach for selecting quarantine nodes in vaccination studies [41]. Betweenness centrality measures the extent to which a node lies on paths between other nodes and is used to detect the amount of influence a particular node has over the flow of information in a graph. The measure is often used to find nodes that serve as critical links between different parts of a graph. Formally, the shortest-path betweenness centrality of a node $i \in V$ on a weighted graph $G$ is

$$g(i) = \sum_{i \neq s \neq t; s, t \in G} \frac{g_{st}(i)}{g_{st}}$$

where $g_{st}$ is the number of shortest paths between nodes $s$, $t$ and $g_{st}(i)$ is the number of geodesic paths between nodes $s$, $t$ that go through node $i$. Both of these measurements calculate path length with respect to the weights of $G$. In words, the betweenness centrality $g(i)$ of a node $i$ is the probability that $i$ lies on a shortest path between some two nodes in $G$. In our camper travel network, higher weights denote more frequent trips, so for the purposes of determining the betweenness centrality, the weight of each edge $(i, j)$ is $max_{i,j}(P_{ij}) + 1 - P_{ij}$.

We model the implementation of firewood quarantine strategies at patches $V$ by introducing a time-dependent term in Eqs (6) and (7). Let $t_0$, and $\Delta t$ be the starting time of the quarantine and the length of the quarantine respectively. Let $H_c(x, \Delta t)$, defined as

$$H_c(x, \Delta t) = \begin{cases} 1 & x < 0 \\ 0 & 0 \leq x \leq \Delta t \\ 1 & x > \Delta t \end{cases}$$

be an upside-down boxcar function of length $\Delta t$. This function acts as a switch which is "off" whenever $x \in [0, \Delta t]$. With this function, we can modify Eqs (6) and (7) so that patches $i \in V$ do not import or export firewood whenever $x \in [0, \Delta t]$.

If $i \in V$,

$$\frac{dI_i}{dt} = -\gamma I_i + AS_i(I_i + B_i)\theta_k(I_i + B_i) - dH_c(t - t_0, \Delta t)\sum_{j=1, j \neq i}^{N} P_{j,i}(1 - C_e)(1 - L_i)I_i \qquad (9)$$

$$\frac{dB_i}{dt} = -\gamma B_i + dH_c(t - t_0, \Delta t)\sum_{j=1, j \neq i}^{N} P_{i,j}(1 - C_e)(1 - L_j)I_j \qquad (10)$$

Note that the only difference in the new Eqs (9) and (10) from Eqs (6) and (7) is in the last term denoting the interactions with neighbouring nodes. The equations for patches not in

under quarantine (i.e., not in $V$) require us to distinguish arcs that connect to and from nodes under quarantine in $V$.

If $i \notin V$,

$$\frac{dI_i}{dt} = -\gamma I_i + AS_i(I_i + B_i)\theta_k(I_i + B_i) \quad - \sum_{j=1, j\neq i, j\notin V}^{N} P_{j,i}(1 - C_e)(1 - L_i)I_i$$

$$-dH_c(t - t_0, \Delta t) \sum_{j=1, j\neq i, j\in V}^{N} P_{j,i}(1 - C_e)(1 - L_i)I_i$$

(11)

$$\frac{dB_i}{dt} = -\gamma B_i + \sum_{j=1, j\neq i, j\notin V}^{N} dP_{i,j}(1 - C_e)(1 - L_j)I_j + dH_c(t - t_0, \Delta t) \sum_{j=1, j\neq i, j\in V}^{N} P_{i,j}(1 - C_e)(1 - L_j)I_j \quad (12)$$

In Eqs (11) and (12) we split the summation term into two summations, one over all patches which are not under quarantine (i.e., not in the set $V$) and patches under quarantine in $V$. The latter summation is multiplied by a term, $H_c(t−t_0, \Delta t)$ which switches on and off the quarantine conditions.

## Parameterization

We used data from [10] and [37], to quantify the risk of firewood transport to recreational destinations in Central Canada. The data documented the movements of campers to provincial campgrounds in Ontario, Quebec and Manitoba. Such data are maintained by provincial ministries of natural resources (MNRs). The dataset included a large number of potential origin sites (i.e., approximately 9000 locations). To reduce the computational burden, we reduced the size of the camper travel network by including all recreational destination locations but considering only the origin locations in the Canadian provinces of Ontario, Manitoba, and Quebec. We further reduced the size of the network by selecting most travelled routes. We selected the largest subgraph with a minimum degree of 10 (the 10-core of the graph) which considered only the most connected nodes, with largest impact on pest transmission. We implemented the procedure using the NetworkX library [42]. The final camper travel network included 2250 sites (Fig 1).

Because the model uses a large camper travel network it has a very large parameter space, and many of the parameters, especially those in Eq 8, are difficult to estimate directly from data. In this study we are exploring the region of parameter space that most closely approximates the dynamics in real infestations, such as the typical size and duration of the recent emerald ash borer outbreak in eastern Canada. To select the most relevant range of the social influence parameters, $\sigma$, $s$, $f$, which are difficult to estimate from the literature, we did sensitivity analyses over a wide range of these parameters, and identified the parameter space where these parameters had the largest effect on the model dynamics, and where the course of the invasion was realistic. The inter-patch and intra-patch infection rate parameters, $d$, $A$, were selected to infest and eventually kill at least 95% of the tree population within 10 to 15 years.

We integrated Eqs (5)–(8) using code written in the Julia language, using the JuliaDiffEq library [43]. The integration was run on the Compute Canada clusters. Our primary focus was to explore the relative impacts of firewood enforcement versus public outreach and their abilities to reduce pest infestation rates across the camper travel network. We consider a hypothetical scenario where a harmful invasive pest is introduced in the largest urban center in eastern Canada with foreign imports (Greater Toronto Area, GTA) and assume that the bulk host tree population in the GTA is infested. This scenario is based on a history of past entries of invasive

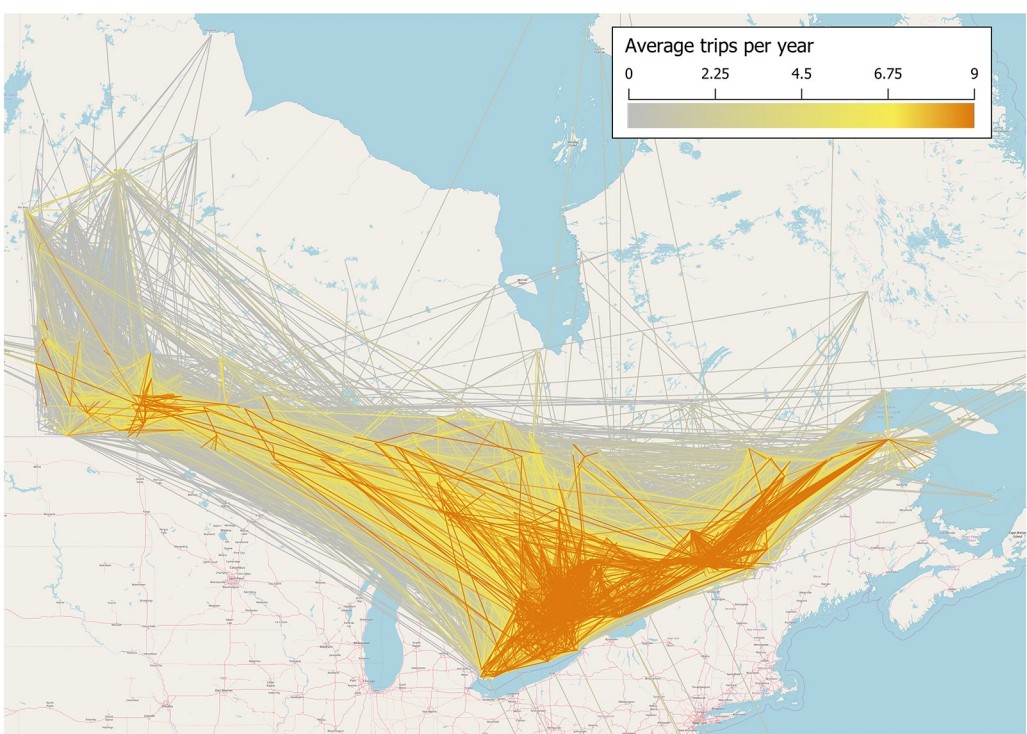

**Fig 1. Camper travel network in Ontario, Quebec and Manitoba.** Darker (more orange) lines represent more trips.

wood-boring pests to the GTA with foreign imports (such as introduction of Asian long-horned beetle in Toronto and Mississauga [44]).

## Assessing intervention efficacy

The primary statistic we use to assess the total mortality of an infestation after $t$ years is the average cumulative infested population, $\frac{1}{N}\sum_{i=1}^{N} T_i(t)$. To calculate $T_i(t)$, the cumulative infested population at patch $i$ and time $t$, we solve the following equation in addition to the model equations.

$$\frac{dT_i}{dt} = AS_i(I_i + B_i)\theta_k(I_i - I_a) \tag{13}$$

The right-hand side of Eq 13 is the only positive term of Eq 6, so it increases when new infested trees are added to $I_i(t)$, but does not decrease when infested trees die, thereby counting the total number of infestations.

Since it is difficult to determine what utility value $U$, which defines the social cost of transport, corresponds to the current level of funding, we try to answer whether it would be beneficial to increase the funding, which we call the marginal benefit of increasing $U$. Given a time $\bar{t}$, we calculate $T(\bar{t})$ for a set of $U \in [-5, 5]$, then we fit a linear function of $U$ to these points. We find a first-order approximation of $T(\bar{t})$ change per unit $U$ (Fig 4) for a given set of parameters and time $\bar{t}$. A positive slope indicates that total infested tree population increases when $U$ is increased, which means that increasing $U$ does not reduce the impact of the pest (at least, to a first approximation). In Figs (4)–(6) this method is used to show how the total number of infested trees changes with respect to an increase in $U$, as a function of parameters and time.

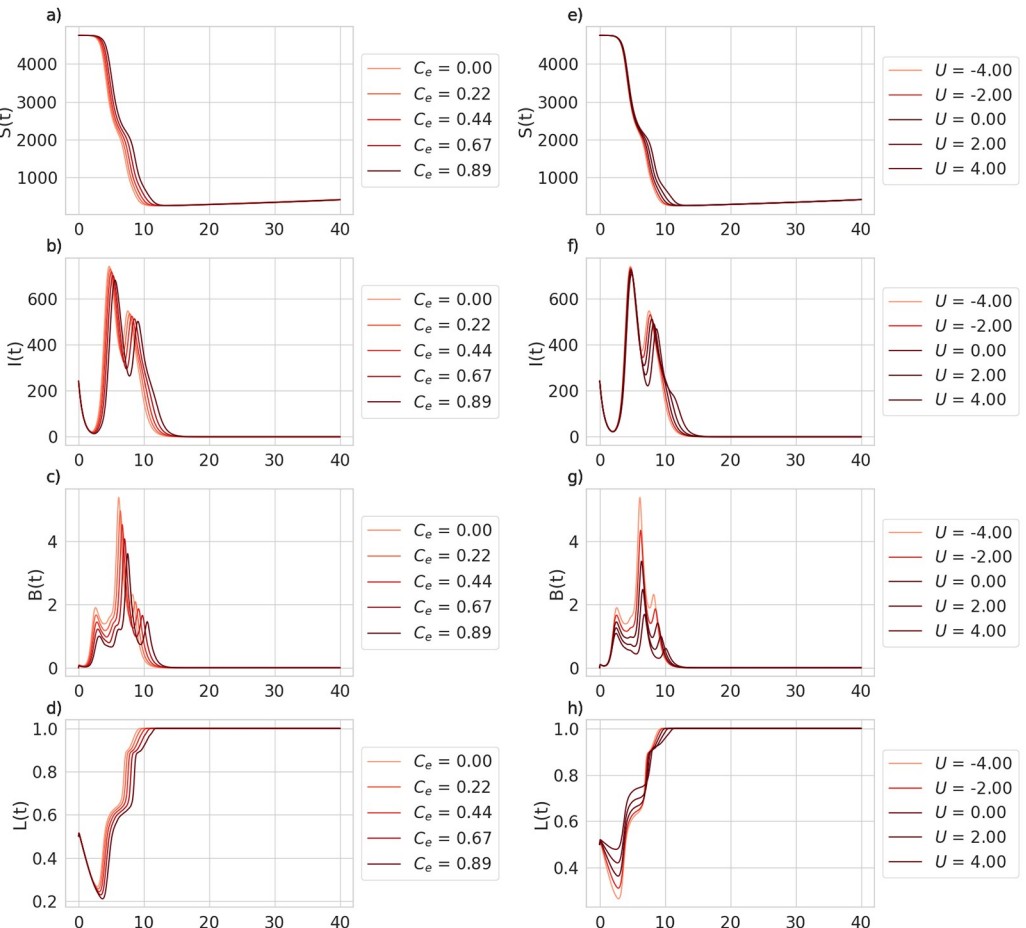

**Fig 2. Time series of model variables as a function of interventions, direct (raising $C_e$, panels a—d) and through social pressure (raising $U$, panels e—h).** The former intervention, panels a-d, means an increase of social pressure on people who choose to transport firewood (i.e. increasing the $U$ value), and the latter refers to direct interception of firewood (i.e. increasing the $C_e$ value). Terms $L(t) = \frac{1}{N}\sum_{i=1}^{N} S_i(t)$, $I(t) = \frac{1}{N}\sum_{i=1}^{N} I_i(t)$, $B(t) = \frac{1}{N}\sum_{i=1}^{N} B_i(t)$, $L(t) = \frac{1}{N}\sum_{i=1}^{N} L_i(t)$ are the averages of the state variables over all patches. $S(t)$ has been omitted for brevity.

## Results

In our baseline scenario (Fig 2, parameters as in Table 1), the model shows a typical pest outbreak originating in the GTA infesting all campgrounds in Ontario, Manitoba and Quebec over 10-20 years. This agrees with the observed timescale of the recent infestation of emerald ash borer (EAB) which entered Ontario in 2002 and now has infested most major populated places in the province [45].

First we discuss the timeseries plot of the baseline parameters (Table 1), where the model variables are averaged over all of the patches for easier visualization (Fig 2). Accordingly, we define $I(t) = \frac{1}{N}\sum_{i=1}^{N} I_i(t)$, $B(t) = \frac{1}{N}\sum_{i=1}^{N} B_i(t)$, $L(t) = \frac{1}{N}\sum_{i=1}^{N} L_i(t)$, to be the average infested tree population at $t$, the average quantity of infested logs at $t$, and the average fraction of local strategists at $t$, respectively. In Fig 2, we find that increasing $U$ (the social cost to transport firewood) increases the number of local strategists $L(t)$ (Fig 2h)—people who choose not to transport firewood between patches— and also reduces the size of the invasion, (Fig 2f) and the average number of infested logs, $B(t)$ (Fig 2g). Although the reduction in B(t) is significant (as

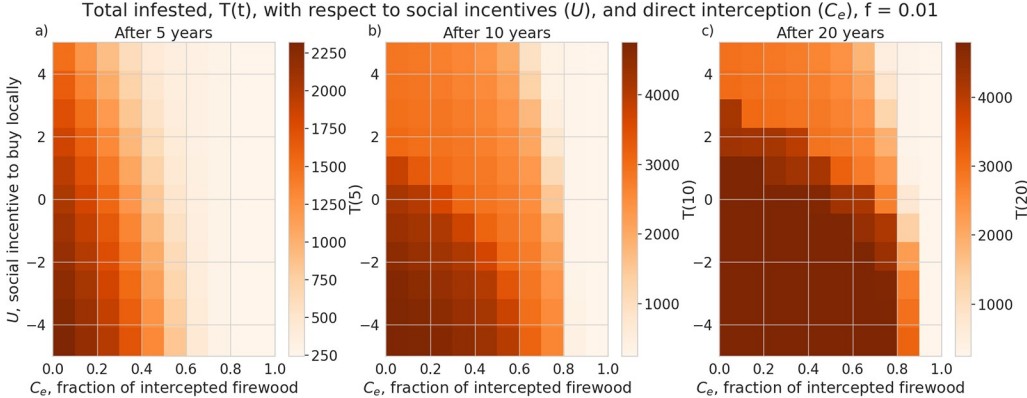

**Fig 3. Total infestation per node over 5, 10 and 20 years.** Neither increasing $U$ nor $C_e$ are effective at long time scales.

shown by the large differences in light red and dark red time series in Fig 2g), the flattening of the curve for infested trees (Fig 2f) is comparatively less significant. We can compare this with the result of increasing the fraction of infested logs intercepted between patches, $C_e$ (2a,b,c,d). Increasing $C_e$ decreases the number of infested trees, the delays the peak of the outbreak (Fig 2b and 2c). The delay in the peak of the outbreak also appears to cause the lag in $L(t)$ (Fig 2d). Social incentives appear to be very effective at reducing $B(t)$ while being less effective at reducing $I(t)$. This indicates that a shift from transport strategists to local strategists primarily occurs in areas that have already been infested. This effect does not occur with direct interception of infested firewood. Notably, direct interception is difficult to implement effectively, as even after intercepting high proportions of the infested wood transport, the corresponding decrease in $I(t)$ remains low (Fig 2b).

In Fig 3 we show the total number of infested trees at time $t$, $T(t)$, with respect to combinations of $U$, the social cost to transport firewood, and the fraction of infested firewood intercepted, $C_e$. If the fraction of intercepted infested firewood, $C_e$, is greater than 80%, we see a sharp reduction in the total infestation, $T$, even after 20 years (Fig 3c), but lower interception rates have little effect unless the social cost to transport $U$ is above the threshold seen in panel c) (Fig 3). Over a shorter time scale, increasing $C_e$ appears to be effective at all interception rates.

The parameter $f$ controls how the proportion of strategists in a given patch $i$ ($L_i(t)$) responds to the population of infested trees ($I_i$) in that patch (Eq 8). Since social incentives (such as an intervention to human-mediated pest transport) tend to be less effective because they prevent firewood transport mostly in the areas that have already been colonized by pests (as suggested in Fig 2), we consider how the parameter $f$ affects the marginal returns on $U$ over time (Fig 4). The shade of the blue region in Fig 4 represents the degree to which increasing $U$ is beneficial, corresponding to a negative slope in the linear approximation of the change in $T$ with respect to $U$ (Fig 4 inset). Similarly, a red cell indicates non-negative slope and therefore a neutral or detrimental marginal effect. We begin to see the benefit of increasing $U$ after about 10 years, shown by the transition from lighter blue to dark blue as we move from the bottom of the image to the top (Fig 4. This relationship is only affected slightly by altering the impact of local infestation on local strategy, $f$, where we begin to see slightly detrimental marginal returns after 10 years if $f < 0.04$.

Similarly, we have compared the marginal returns on increasing $U$ with respect to the intra-patch transmission rate $A$ and time $t$ (Fig 5). When $A$ is small ($A \leq 0.0009$, beneficial marginal returns on $U$ can be observed over the whole duration of the infestation. We further explore

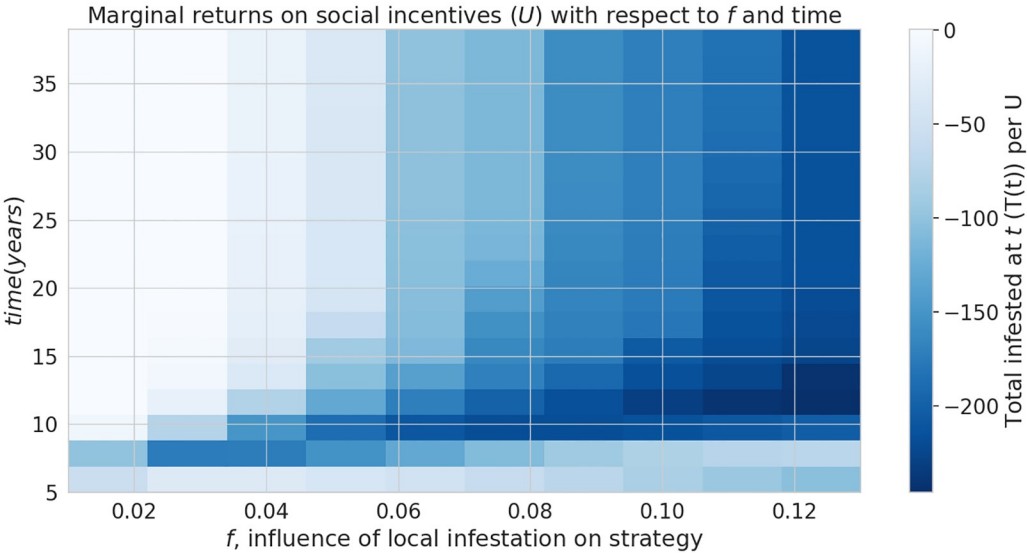

**Fig 4. Efficacy of social incentives on infestation after time *T*.** Inset graph shows an example of cross-section along the line *f* = 0.11 The influence of infestation on transport strategy, *f*, can hinder the intervention by public outreach, in the long-term (after approximately 20 years). The inset figure illustrates how one column in the heat map, shown by the dotted line, is constructed from the slopes of linear approximations of *T*(*t*) over *U* ∈ [−5, 5]. The blueness of the lines going left to right is a function of their slope, corresponding to the color of the cells in the heatmap.

the impact of varying the rate of transmission of infested firewood between patches, *d* (Fig 6). We find a roughly parabola-shaped region in the parameter plane of intra-patch and inter-patch transmission rates (*A* and *d* respectively), above which the marginal returns of increasing *U* are zero or possibly detrimental to the size of the total infested population after 10-20 years. Larger intra-patch transmission rates enable the pest population to establish earlier in a given patch by propagules. We see good marginal return in parameter regimes where few

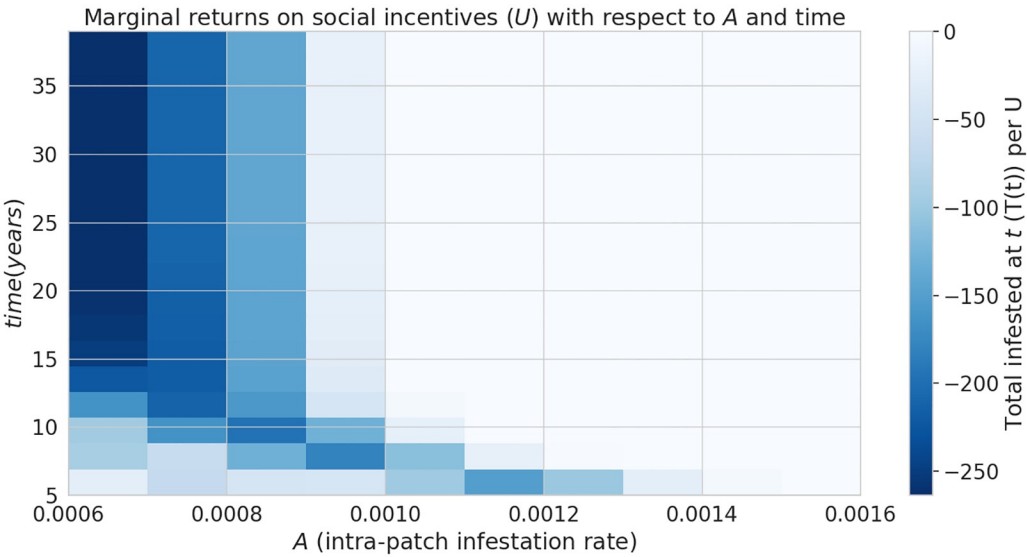

**Fig 5. Efficacy of social incentives on infestation after time period *T* with respect to *A*, the intra-patch infestation parameter.** This intervention becomes ineffective over time if *A* is sufficiently large.

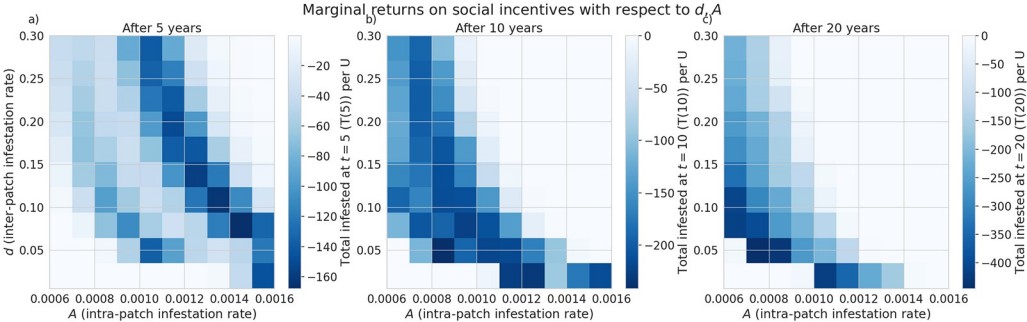

**Fig 6. Efficacy of social incentives on infestation after time $T$ intra-patch spreading rate $A$, affects infestation outcomes.** The social incentive to not transport firewood, $U$, is more effective with lower pest spread rates.

transport strategists (high $L(t)$) would reduce the reproductive ratio of the infection below 1. For instance, at the point $(A, d) = (0.00126, 0.103)$, increasing $U$ is able to delay and eventually prevent a second wave, which decreases the total number of infected trees significantly (S1 Fig). If the transmission rates $A$, $d$ are high enough that even with no transport strategists, we get a second wave of infection, the effect of increasing $U$ can be slightly detrimental (S2 Fig). Panel f) of the aforementioned figures plots the number of patches where $I \geq 1$ over time, showing that the detrimental effect is largely due to the infection persisting longer in the network.

We also explored the effectiveness of patch quarantine by replacing model Eqs (6) and (7) with Eqs (9)–(12). This replacement prevents individual patches (nodes in a set $V$) with the highest betweenness centrality (with respect to the weights $P_{ij}$) from interacting with their neighbours during the time of the quarantine ($t \in [t_0, t_0 + \Delta t]$). Imposing quarantine on these nodes is expected to have the greatest impact on pest transmission rate. If the quarantine is initiated one year after the pest is introduced into the system (that is, $t_0 = 1.0$) then we find a significant reduction in total infestation even if only 50 patches are quarantined ($|V| = 50$) assuming they are quarantined for more than a year, shown in Fig 7. However, in our model, we find that quarantines need to be longer than approximately three years, and involve more than 150 nodes to still be effective in reducing the total infested population after 20 years

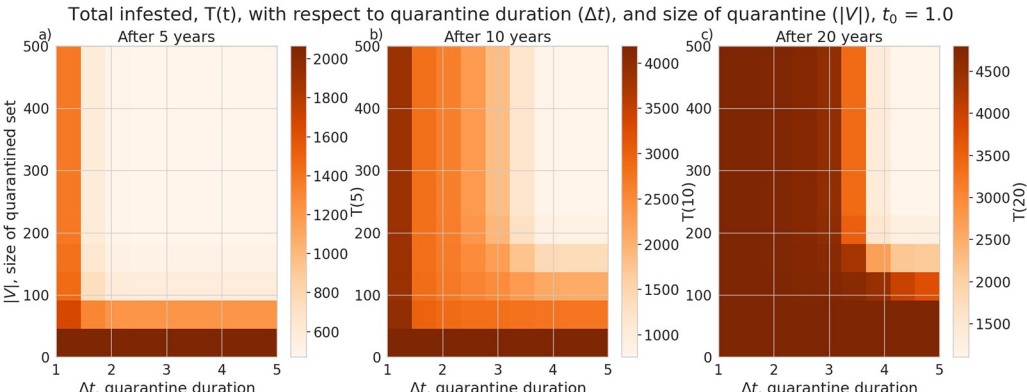

**Fig 7. Average total infested trees ($T(t)$) after 5, 10 and 15 years (panels a),b), and c) respectively), assuming the quarantine begins one year after the pest is introduced.** Total infestation plotted with respect to the number of nodes quarantined ($|V|$) and the length of the quarantine ($\Delta t$). The quarantine is effective over 5 years with only 50 patches, provided they are closed for over a year.

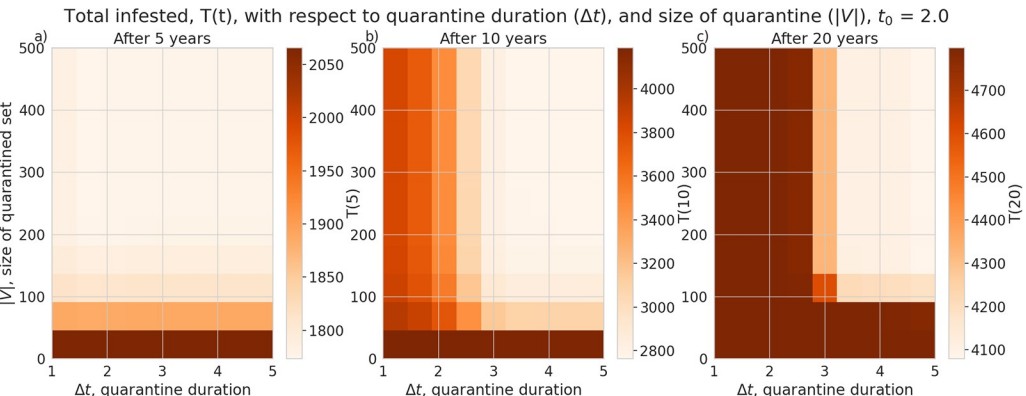

**Fig 8. Average total infested trees ($T(t)$) after 5, 10 and 15 years (panels a),b), and c) respectively), assuming the quarantine begins two years after the pest is introduced.** Total infestation plotted with respect to the number of nodes quarantined ($|V|$) and the length of the quarantine ($\Delta t$).

$T(20)$. An interesting result in our quarantine plots is that we see a slightly larger range of effective parameter values if the quarantine begins after two years, $t_0 = 2.0$ (Fig 8), rather than one, $t_0 = 1.0$. This effect is probably due to the delay in infestation after the model is initialized, which can be seen by the local minimum in the infestation timeseries (Fig 2b and 2f).

## Conclusion

We presented a model coupling human social behaviour regarding transport of infested firewood through recreational travel with a model of the spread of an invasive forest pest. Our main focus was to compare, in relative terms, common measures for slowing the spread of invasive species with firewood transport, such as public outreach campaigns aimed to raise awareness about the problem, and enforcement measures, including inspections at checkpoints to control the movement of firewood, and location-specific quarantine. The model is parameterized with campground reservation data for provincial parks and campgrounds in the provinces of Ontario, Manitoba and Quebec, Canada and incorporated spatial information on the topology and geographical configuration of the camper travel network.

Under the assumptions of our model and a particular camper travel network configuration used in our model, checkpoints to control the movement of untreated firewood are unlikely to be effective at slowing the spread of invasive forest pests with firewood transport given typical moderate levels of funding and long delays in the response measures. We find the rate of interception to halve the total infested tree population after 5 years is about 30% (Fig 3), which is unlikely to be achieved in practice given typical limited budgets and personnel constraints in present-day firewood control programs. Given that our model uses somewhat simplified assumptions and does not account for fine-scale logistical constraints (which are inspectors may face in various spatial locations) the actual rate of interception is likely to be lower in practical conditions. While a previous study [18] that used a similar model has demonstrated that social incentives may improve outcomes in a two-patch model under equilibrium conditions, we have found that in our complex landscape network, the outcomes of infestation and invasion control measures are highly dependent on the time scale and the characteristics of the invaders, such as the inter-patch and intra-patch infestation rate. Social incentives (which aim to decrease the transport of firewood, $U$), are generally able to reduce the infestation rate in the short term but its effectiveness is highly dependent on the ability of the pest to spread and infest other locations (Figs 5 and 6) under the conditions we have

explored. Humans in our model tend to reduce their transport of firewood between patches in already infested areas, which causes the pest to persist longer in the network (Fig 2). Our results show that there could exist a threshold in the pest transmission rate $A$ and the proportion of the infested wood which is turned into firewood, $d$ (Fig 6). Below this threshold, it would not be beneficial to increase social outreach (i.e., increase $U$). This insight could be helpful in determining the spatial allocation of firewood movement control efforts for a particular pest species. We have also found that the location-specific quarantines that aim to restrict the movement of firewood to/from a particular location, might only be effective at slowing the invasion spread if a sufficiently large number (at least 140 in our case) of highly connected locations is quarantined, and the quarantine is established at early stages of infestation (Figs 7 and 8).

Given the typical cost limitations and logistics constraints faced by today's firewood control programs, and the assumptions made in our modeling framework, it is unlikely that local quarantine measures could significantly slow the spread of invasive pests through firewood unless drastic control and quarantine measures are undertaken. Public outreach campaigns, while helping increasing awareness of problem, cannot reliably slow the spread of pests within the parameter values tested, when the invasion spreads through a network based on camper travel data in Manitoba, Ontario and Quebec. Within our model, public outreach could be more effective for slow-spreading pests when the organism is able to kill host trees quickly but does not have significant spread capacity (that is, the inter-patch and intra-patch infestation rates are sufficiently small). Direct intervention, such as checkpoint inspections for illegally transported firewood, is also not an option, because meaningful outcomes can only be achieved if significant fractions of firewood transports can be intercepted. We find that patch quarantine is effective at slowing, or even stopping, the spread of an invasive forest pest when a large number of highly-connected patches are quarantined, for a long enough period. Our results in general terms agree with a present-day situation when numerous outreach and local quarantine measures had limited impact on illegal transport of firewood by campers and failed to slow the spread of wood-boring pests transported with untreated firewood. Our results also indicate that the enforcement campaigns aimed to intercept illegal movement of untreated firewood can only be effective if implemented at very large spatial scales in timely fashion (which, in turn, would require massive amounts of funding and personnel support).

There are some shortcomings to our model that could be addressed in future work. The interventions we study do not have spatial or time specifications for individual locations in the camper travel network. Deciding where and when, to deploy the outreach and enforcement measures in a particular location would be a major enhancement of the model. Second, our model depicted a general problem of an invasive pest spreading with untreated firewood moved by recreational travelers. To adapt the problem to a particular pest species, a more specialized spread model will be required. We simplified the model by assuming that each infested patch provides similar propagule pressure to recreational travellers leaving the infested site. This assumption was made because no data about the actual proportions of infested wood carried by recreational travellers leaving the infested sites were available. Also, our analysis did not offer much insight at the level of individual spatial locations in a camper travel network. A simpler mechanistic model that applies unique pest control decisions at individual spatial locations could potentially address that aspect. Another possible way to simplify the model would be to remove the tree growth dynamics —since it operates on a longer time scale than the infestation spread— and so an invasion model without the forest growth component could be a reasonable approximation for short-term planning horizons. This will be the focus of future efforts.

## Supporting information

**S1 Fig. Time evolution of model variables for various values of *U* where (*A*, *d*) = (0.0009, 0.038).**
(PDF)

**S2 Fig. Time evolution of model variables for various values of *U* where (*A*, *d*) = (0.00126, 0.103).**
(PDF)

## Acknowledgments

The authors would like to thank Dr. Hanno Seebens and an anonymous reviewer for their contributions. Their detailed and thorough suggestions have significantly improved the quality of our paper.

## Author Contributions

**Conceptualization:** Denys Yemshanov, Madhur Anand.

**Data curation:** Peter C. Jentsch, Denys Yemshanov.

**Formal analysis:** Peter C. Jentsch.

**Funding acquisition:** Chris T. Bauch, Denys Yemshanov, Madhur Anand.

**Investigation:** Peter C. Jentsch, Chris T. Bauch, Denys Yemshanov, Madhur Anand.

**Methodology:** Peter C. Jentsch, Chris T. Bauch, Denys Yemshanov, Madhur Anand.

**Project administration:** Peter C. Jentsch, Chris T. Bauch, Denys Yemshanov, Madhur Anand.

**Resources:** Chris T. Bauch, Denys Yemshanov, Madhur Anand.

**Software:** Peter C. Jentsch.

**Supervision:** Chris T. Bauch, Denys Yemshanov, Madhur Anand.

**Validation:** Peter C. Jentsch, Chris T. Bauch, Denys Yemshanov, Madhur Anand.

**Visualization:** Peter C. Jentsch, Chris T. Bauch.

**Writing – original draft:** Peter C. Jentsch.

**Writing – review & editing:** Peter C. Jentsch, Chris T. Bauch, Denys Yemshanov, Madhur Anand.

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
