## [Decision Letter · Decision Letter 0]

26 Feb 2020

PONE-D-20-00702

Go big or go home: a model-based assessment of general strategies to slow the spread of forest pests via infested firewood

PLOS ONE

Dear Mr. Jentsch,

Thank you for submitting your manuscript to PLOS ONE. After careful consideration, we feel that it has merit but does not fully meet PLOS ONE’s publication criteria as it currently stands. Therefore, we invite you to submit a revised version of the manuscript that addresses the points raised during the review process.

In particular, both reviewers reported a serious lack of clarity in the presentation of the study, figures and supplementary materials, at the point that it is difficult to evaluate the overall quality of the work.

The two reviewers also provided a number of specific comments which I recommend you to address carefully, in the case you wish to resubmit your manuscript.

We would appreciate receiving your revised manuscript by Apr 11 2020 11:59PM. To enhance the reproducibility of your results, we recommend that if applicable you deposit your laboratory protocols in protocols.io, where a protocol can be assigned its own identifier (DOI) such that it can be cited independently in the future. For instructions see: http://journals.plos.org/plosone/s/submission-guidelines#loc-laboratory-protocols

We look forward to receiving your revised manuscript.

Kind regards,

Agnese Marchini

Academic Editor

PLOS ONE

Journal Requirements:

1. Thank you for including your funding statement; "This work has been partially funded by Canadian Forest Service - Canadian Food Inspection Agency Forest Invasive Alien Species Initiative, https://www.nrcan.gc.ca/our-natural-resources/forests-forestry/13497. The funders had no role in study design, data collection and analysis, decision to publish, or preparation of the manuscript."

Reviewers' comments:

Reviewer's Responses to Questions

**Comments to the Author**

1. Is the manuscript technically sound, and do the data support the conclusions?

Reviewer #1: Partly

Reviewer #2: Partly

2. Has the statistical analysis been performed appropriately and rigorously? 

Reviewer #1: Yes

Reviewer #2: I Don't Know

3. Have the authors made all data underlying the findings in their manuscript fully available?

Reviewer #1: No

Reviewer #2: Yes

4. Is the manuscript presented in an intelligible fashion and written in standard English?

Reviewer #1: Yes

Reviewer #2: Yes

5. Review Comments to the Author

Reviewer #1: Jentsch et al. present a modelling study to investigate the spread of forest pests through the transport of firewood by campers in Canada. They combined and refined existing models to simulate the interaction between the population dynamics of host trees, the spread of forest pests by campers and human behaviour in following different strategies of accessing firewood. The ultimate goal of the study was to test the effectiveness of different strategies to mitigate the spread of forest pests by controlling firewood, public education or quarantine sites.

My overall impression is that the modelling part of study was thoughtfully designed, while less attention has been paid to the presentation of results and their interpretation. Consequently, while the methodology and experiments seemed to be appropriate, the conclusions drawn by the authors are not fully supported by the study results, which, however, could be addressed by rephrasing the text. I therefore do not have any major comment to the study design, but I had some difficulties to understand the study results and did not completely agree with the conclusions drawn by the authors. I would very much appreciate if the authors put as much as efforts in presenting their results as compared to the presentation of the model, which is much clearer. I list my major comments first, followed by minor ones:

I had a hard time to understand the study, and I think that I did not fully get the story in all details for different reasons:

1. The presentation of results is very complex with an overwhelming number of figures, many of them are not discussed and not even mentioned in the text. It would help a lot to reduce the number of figures to those, which are really necessary to understand the main story, and to clearly explain what is shown and how to interpret it.

2. It was often difficult to understand the text particularly when the authors used variable names instead of textual description. For instance, sentences like “We illustrate how the marginal return of the U values varies over time using the method described in section (Figs 4, 5).” (l. 299) are nearly impossible to understand. What is meant by “marginal return”? Please do not only mention the variable name (“U values”) as by reading the text the first time, the reader usually do not remember the meaning of the variable names. It requires to always go back-and-forth to check their meanings. Which method do you refer to? Please name it instead of referring to some section or figures.

3. The interpretation of the figures are hard to follow. For instance, in line 300 Figs 4,5 are mentioned, but I did not understand where the following lines (300-308) refer to. I first thought Fig. 4 but this seemingly was not the case.

4. Please be more clear about your interpretation. I often did not understand where the authors draw their conclusion from. For instance, “tends to decrease over time” (l. 303) – where can I see this? Or “tend to worsen in the long term” (l. 360) – where did you get this from? Or “The rate of

interception that could significantly reduce the infestation rate must be very high (i.e., > 25%)” - again, how did you get to this value and the conclusion? It is very likely that I overlooked something, but this will certainly happen to others readers as well.

5. The figures were not embedded in the text and not numbered in the manuscript which I had. It was therefore difficult to understand to which figure the authors referred. This may be caused by the journal but added up to the challenge in understanding the study.

6. The model is overly complex, which was also briefly mentioned by the authors. Without having tested the model, it appears to me that the simulation of susceptible trees would not be required and even the detailed simulation of infested trees could be simplified as it is mostly a continues decay over time. But do not misunderstand me: There is no need to change the model, but I wanted to mention it as it contributes to the complexity of the study and makes understanding the results more challenging.

Please pay particular attention to a clear presentation of results throughout the manuscript and help the reader to follow your interpretation of results. Right now this is often very challenging.

Risk of over-selling:

The authors main conclusions are pretty strong in emphasising that several options of mitigation did not work out in a realistic context, but I have some doubts that this can be indeed said based on study results, and I recommend to provide the study results in a more balanced way. One of the authors main conclusion is that none of the tested mitigation strategies can effectively stop the spread of the forest pests (l. 372). I would argue that actually the model is not capable of simulating a stop of the spread as chances of spread are always above zero and trees cannot recover (100% mortality). Consequently, the spread will always continue except drastic countermeasures were put in place such as stopping (nearly) whole movement of campers or cutting all trees. The authors themselves raised the issue that the model may require some refinements to capture realistic dynamics, but then the authors should be more careful with the main conclusions. I simply cannot believe that “quarantines […] appear to be ineffective”. This also depends on the extent of the quarantine as a whole area quarantine should be very effective in halting the spread. Please revised very carefully the whole conclusion and the abstract and present results and conclusions in more balanced way supported by study results and in the context of the capabilities of the model. This would avoid getting the impression of over-selling study results.

Availability of materials:

According to the policy of PlosOne, material to reproduce the study have to be made available by the authors. From the data policy of PlosOne: “Materials - We expect that all researchers submitting to PLOS will make all relevant materials that may be reasonably requested by others available without restrictions upon publication of the work. Software - We expect that all researchers submitting to PLOS submissions in which software is the central part of the manuscript will make all relevant software available without restrictions upon publication of the work. Authors must ensure that software remains usable over time regardless of versions or upgrades.” This is certainly not the case here.

Minor comments:

Abstract: What are “social costs”? Please explained

Line 60: Remove “.” after “.”.

Line 113: Something seems to be missing after “section “.

Line 133: “population of host trees” is not full true as it would include infested trees I as well. Maybe write “population of susceptible trees” instead. And please put the variable name when the term is introduced, e.g. “population of susceptible trees, S”.

Section: Pest spread model: Refer to table 1 early on to make clear that an overview of variables is available.

Line 137: “where I_a determines the smallest viable population of pest” Seems incomplete. In addition, I denotes the infested tree population rather than the pest, right? Please clarify.

Line 162: What are “local strategists” and “transport strategists”? Please explain or avoid these terms.

Table 1: V and N are missing in this table.

Lines 254-257: I did not understand what was done here. What is the “infested category”? Here, it would be helpful to use the actual variable names. And please clarify how it was calculated and what was plotted. Maybe provide an example of the plot in the supplement.

Line 264: Remove “4.”

Results: The first paragraph is still part of the methods. Please move to method sections.

Lines 280-282: I cannot see what is described here in Fig. 2b. Please clarify. In addition, explain Fig. 2 in total or remove panels from the figure to avoid confusion.

Lines 297-298: “Fig 3 also shows that it is always better to use both social interventions U” Where did you see this? Please rephrase and clarify how you got to this conclusion.

Lines 299-308: As already mentioned above this paragraph is also difficult to understand. Please make clear where you get this information from and why did you come to these interpretations.

In general, please carefully revised the whole results section to clarify where the results were taken from and how did you come to the respective conclusions. In addition to the points raised above, there are many minor issues throughout the whole text.

Conclusions: As mentioned above, conclusions should be presented in a more balanced way. And it should always be clear how the conclusion is supported by study results. For instance, I did not understand which result support this conclusion: “In particular, social incentives (which corresponds to increasing the incentive not to transport firewood, U in our model) generally help slightly reduce the infestation rate in the short term but tend to worsen it in the long term.” It is even hard to imagine how social incentives could worsen it (the spread?) in the long term. If the authors present such strong messages, these need be based on solid ground and well supported by the study, which I currently do not see. The same applies to e.g. lines 371-372, 376-377, 387, etc. Please carefully check the whole conclusions.

Supplement: The supplementary material contains ridiculously long lists of numbers without any explanation. Presenting data in this way is pretty useless and contradicts all kinds of data management policies. I strongly suggest to pay more attention to the presentation of results and materials including descriptions of content and meta-data.

Reviewer #2: Please see attachment.

6. PLOS authors have the option to publish the peer review history of their article (what does this mean?). If published, this will include your full peer review and any attached files.

Reviewer #1: No

Reviewer #2: No

---

## [Author Response · Author response to Decision Letter 0]

12 May 2020

We would like to thank the editor and reviewers for their helpful comments and valuable suggestions. Please see our replies, in italics, to comments in the attached response PDFs. We a did a complete rerun of the model simulations in order to address your comments, and in the process, found a bug in the model code which has altered the results in some places, but overall the message of the paper remains the same.

The model data and code have been included as a zip file, "model_data_and_code.zip". This archive also contains a short text file explaining the workflow and a version of the manuscript with figures left inline, which hopefully helps readability.

---

## [Decision Letter · Decision Letter 1]

15 Jun 2020

PONE-D-20-00702R1

Go big or go home: a model-based assessment of general strategies to slow the spread of forest pests via infested firewood

PLOS ONE

Dear Dr. Jentsch,

Thank you for submitting your manuscript to PLOS ONE. After careful consideration, we feel that it has merit but does not fully meet PLOS ONE’s publication criteria as it currently stands. Therefore, we invite you to submit a revised version of the manuscript that addresses the points raised during the review process.

We look forward to receiving your revised manuscript.

Kind regards,

Agnese Marchini

Academic Editor

PLOS ONE

Additional Editor Comments (if provided):

The revised manuscript has addressed the major concerns raised by the reviewers, and the new version is clearer.

However, there are aspects still needing attention. I especially encourage the authors to justify parameter choice.

Reviewers' comments:

Reviewer's Responses to Questions

**Comments to the Author**

1. If the authors have adequately addressed your comments raised in a previous round of review and you feel that this manuscript is now acceptable for publication, you may indicate that here to bypass the “Comments to the Author” section, enter your conflict of interest statement in the “Confidential to Editor” section, and submit your "Accept" recommendation.

Reviewer #1: All comments have been addressed

Reviewer #2: (No Response)

2. Is the manuscript technically sound, and do the data support the conclusions?

Reviewer #1: Yes

Reviewer #2: Partly

3. Has the statistical analysis been performed appropriately and rigorously? 

Reviewer #1: Yes

Reviewer #2: N/A

4. Have the authors made all data underlying the findings in their manuscript fully available?

Reviewer #1: Yes

Reviewer #2: Yes

5. Is the manuscript presented in an intelligible fashion and written in standard English?

Reviewer #1: (No Response)

Reviewer #2: Yes

6. Review Comments to the Author

Reviewer #1: The authors have done a good job in revising the whole manuscript. One of my major concerns in the first round was the confusing presentation of results, which the authors addressed appropriately. It is still not an easy task to fully get the results, but this is mostly due to the nature of the study. But the authors modified the presentation of results now allowing the reader to more easily follow the author’s description. I was also happy to see that the interpretation of the results is now toned down and done in a more balanced way and in line with the study results.

I only have two very minor point left:

In the data availability statement, the authors only state that “all data are fully available without restriction”, but it requires to know where to find the data (and code?). Maybe I missed it.

If I assigned the correct figure to Fig. 2, it seems that panel d and h are not described in the legend, right?

Reviewer #2: Please see attachment. This requires a moderate revision. .

7. PLOS authors have the option to publish the peer review history of their article (what does this mean?). If published, this will include your full peer review and any attached files.

Reviewer #1: Yes: Hanno Seebens

Reviewer #2: No

---

## [Author Response · Author response to Decision Letter 1]

28 Jul 2020

Responses to the reviewer comments are available in the files "Plos_reviewer_response_1.pdf" and "Plos_reviewer_response_2.pdf", respectively.

Thank you

---

## [Editor Report · Decision Letter 2]

6 Aug 2020

PONE-D-20-00702R2

Go big or go home: a model-based assessment of general strategies to slow the spread of forest pests via infested firewood

PLOS ONE

Dear Dr. Jentsch,

Thank you for submitting your manuscript to PLOS ONE. After careful consideration, we feel that it has merit but does not fully meet PLOS ONE’s publication criteria as it currently stands. Therefore, we invite you to submit a revised version of the manuscript that addresses the points raised during the review process.

We look forward to receiving your revised manuscript.

Kind regards,

Agnese Marchini

Academic Editor

PLOS ONE

Additional Editor Comments:

There are two minor, technical aspects to be fixed.

1) It would be kind to thank the reviewers in an Acknowledgement section, for the very careful work they have done. In particular, one of the reviewers has undisclosed his identity (see Decision letter of June 15), and in this case he should be personally acknowledged.

2) There are errors in the reference of Figures: "??" appears instead of the figure number.

Please amend these and resubmit.

---

## [Author Response · Author response to Decision Letter 2]

7 Aug 2020

Thank you for the comments. Here are our responses to the editor comments. 

There are two minor, technical aspects to be fixed.

1) It would be kind to thank the reviewers in an Acknowledgement section, for the very careful work they have done. In particular, one of the reviewers has undisclosed his identity (see Decision letter of June 15), and in this case he should be personally acknowledged.

We have added an acknowledgements section, thank you for the suggestion.

2) There are errors in the reference of Figures: "??" appears instead of the figure number.

This has been addressed.

---

## [Editor Report · Decision Letter 3]

28 Aug 2020

Go big or go home: a model-based assessment of general strategies to slow the spread of forest pests via infested firewood

PONE-D-20-00702R3

Dear Dr. Jentsch,

We’re pleased to inform you that your manuscript has been judged scientifically suitable for publication and will be formally accepted for publication once it meets all outstanding technical requirements.

Kind regards,

Agnese Marchini

Academic Editor

PLOS ONE
---

## [Editor Report · Acceptance letter]

4 Sep 2020

PONE-D-20-00702R3 

Go big or go home: a model-based assessment of general strategies to slow the spread of forest pests via infested firewood 

Dear Dr. Jentsch:

I'm pleased to inform you that your manuscript has been deemed suitable for publication in PLOS ONE. Congratulations! Your manuscript is now with our production department. 

Kind regards, 

on behalf of

Dr. Agnese Marchini 

Academic Editor

PLOS ONE